# RESTRAIN: FROM SPURIOUS VOTES TO SIGNALS — SELF-DRIVEN RL WITH SELF-PENALIZATION

**Zhaoning Yu**[†,1,4,*], **Will Su**[†,4], **Leitian Tao**[3], **Haozhu Wang**[2],
**Aashu Singh**[4], **Hanchao Yu**[4], **Hongyang Gao**[1], **Jianyu Wang**[4],
**Weizhe Yuan**[2,5], **Jason Weston**[2,5], **Ping Yu**[‡,2], **Jing Xu**[‡,2]

[1]Iowa State U    [2]FAIR at Meta SuperIntelligence Lab    [3]UW–Madison    [4]Meta    [5]NYU

## ABSTRACT

Reinforcement learning with human-annotated data has boosted chain-of-thought reasoning in large reasoning models, but these gains come at high costs in labeled data while faltering on harder tasks. A natural next step is experience-driven learning, where models improve without curated labels by adapting to unlabeled data. We introduce REinforcement learning with Self-resTRAINt training (**RESTRAIN**), a self-penalizing RL framework that converts the absence of gold labels into a useful learning signal. Instead of overcommitting to spurious majority votes, RESTRAIN exploits signals from the model's entire answer distribution: penalizing overconfident rollouts and low-consistency examples while preserving promising reasoning chains. This self-penalization mechanism integrates seamlessly into policy optimization methods such as GRPO, enabling continual self-improvement without supervision. On challenging reasoning benchmarks, RESTRAIN delivers large gains using only unlabeled data. With Qwen3-4B-Base and OctoThinker Hybrid-8B-Base, it boosts Pass@1 by up to **+140.7% on AIME25**, **+36.2% on MMLU_STEM**, and **+19.6% on GPQA-Diamond**, nearly matching gold-label training while using no gold labels. These results demonstrate that RESTRAIN establishes a scalable path toward stronger reasoning without gold labels.

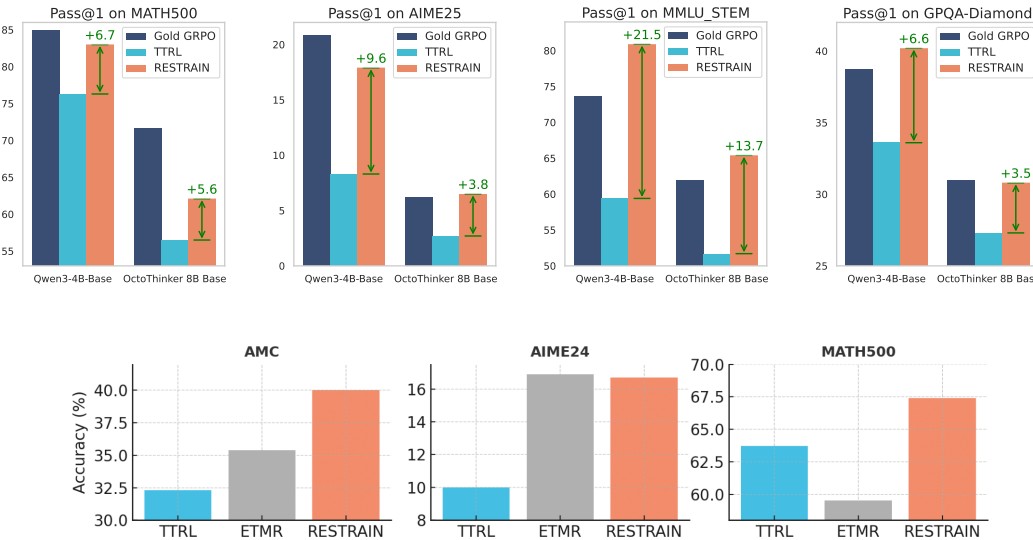

Figure 1: **Performance of Label-free and Test-Time RL.** Top: Pass@1 of Qwen3-4B-Base and OctoThinker Hybrid-8B-Base trained on DAPO-14k-MATH. RESTRAIN outperforms TTRL and nearly matches the Gold-label GRPO upper bound, even surpassing it on MMLU-STEM and GPQA-Diamond. Bottom: Test-time training Llama3.1-8B-Instruct using unlabeled test data from AIME24, AMC23, and MATH500, reporting Pass@1 accuracy. RESTRAIN significantly outperforms TTRL and ETMR, especially on AMC and MATH500.

---

*Work done during an internship at Meta

†Joint first author    ‡Joint last author

# 1 INTRODUCTION

Recent advances in LLMs (Guo et al., 2025; Jaech et al., 2024; Yang et al., 2025) show that Reinforcement Learning (RL) with human-annotated data and verifiable rewards (RLVR) greatly enhances long chain-of-thought reasoning (Wei et al., 2022), achieving strong performance on challenging benchmarks. Yet RLVR remains limited: it depends on ever-growing quantities of high-quality labeled data. Achieving superhuman performance, models will eventually need to operate in environments where even humans lack definitive answers and cannot offer reliable feedback on outputs. In these situations, models must develop the ability to self-improve without direct supervision. This motivates exploring RL on unlabeled data, where progress arises from self-improvement rather than curated labels, with large external corpora serving as a training signal (Zuo et al., 2025). In this work, we study RL in an unsupervised setting to advance reasoning generalization.

A central challenge in enabling self-improvement without labeled data is how a model can generate its own learning signals. One natural direction is self-rewarding methods, where the model generates its own reward signals—for instance, ranking or scoring its rollouts based on its own judgments (Yuan et al., 2024). While these methods remove the dependence on gold labels, evidence remains limited that such methods consistently improve performance on complex reasoning tasks. A second line of work leverages the model's internal agreement, such as using majority voting across multiple rollouts (Zuo et al., 2025; Shafayat et al., 2025; Liu et al., 2025a; Prasad et al., 2024). Yet this approach suffers from reliability and robustness issues that can cause model training collapse: models frequently generate responses with low self-consistency or low confidence across multiple attempts, and for challenging reasoning tasks, the majority-voted answer itself can be systematically flawed. In such cases, minority rollouts can contain the correct solution (Stahlberg & Byrne, 2019; Stahlberg et al., 2022), but these are ignored when overconfident spurious majorities dominate. Training on such distorted reward signals limits scalability as task diversity and complexity increase. The key challenge, therefore, is not merely generating self-derived rewards, but ensuring that they provide robust signals that drive genuine reasoning improvement.

To address this gap, we introduce RESTRAIN, a framework for self-driven RL with self-penalization. Instead of relying on gold labels or external supervision, RESTRAIN leverages the model's own predictions by (1) considering all predicted answers rather than only majority votes, (2) penalizing low-confidence rollouts with negative advantages, and (3) down-weighting low-agreement prompts with fragile majority votes. By integrating self-penalization directly into the RL objective, RESTRAIN turns the absence of labels into rollout-level and prompt-level learning signals. We evaluate RESTRAIN on two base models and two tasks across six benchmarks. Notably, RESTRAIN raises Pass@1 by 140.7% on AIME25, 36.2% on MMLU_STEM, and 19.6% on GPQA-Diamond. Even more striking, its performance nearly matches gold-label supervision—lagging by only 0.4 points. These results establish RESTRAIN as a scalable approach to self-driven RL, pushing reasoning models beyond supervised limits.

# 2 RESTRAIN

We introduce the main ideas of RESTRAIN below and in Figure 3.

**Preliminaries** We adopt Grouped Relative Policy Optimization (GRPO) (Shao et al., 2024) as our main RL algorithm. GRPO optimizes a policy $\pi_\theta$ by sampling $n$ rollouts per prompt $x$ with gold label $y$, and updating with a PPO-style objective against a fixed reference policy $\pi_{\text{ref}}$, using a group-mean baseline for variance reduction. For each rollout $y_i$, we denote by reward $r_i = R(y_i, y|x)$ with advantage $A_i$. The GRPO objective for each prompt $x$ with gold label $y$ is:

$$\mathcal{L}_{\text{GRPO}}(x, y; \theta) = \frac{1}{n} \sum_{i=1}^{n} \min\Big(\rho_i(\theta)\, A_i,\ \text{clip}\big(\rho_i(\theta), 1 - \epsilon, 1 + \epsilon\big) A_i\Big)\ -\ \beta \mathbb{D}_{KL}\left[\pi_\theta \| \pi_{\text{ref}}\right] \quad (1)$$

## 2.1 PSEUDO-LABEL WEIGHTING

In unsupervised settings without gold labels, a model can give multiple predictions for a given prompt $x$, regardless of their correctness. Figure 2 reports accuracies on model predicted answers

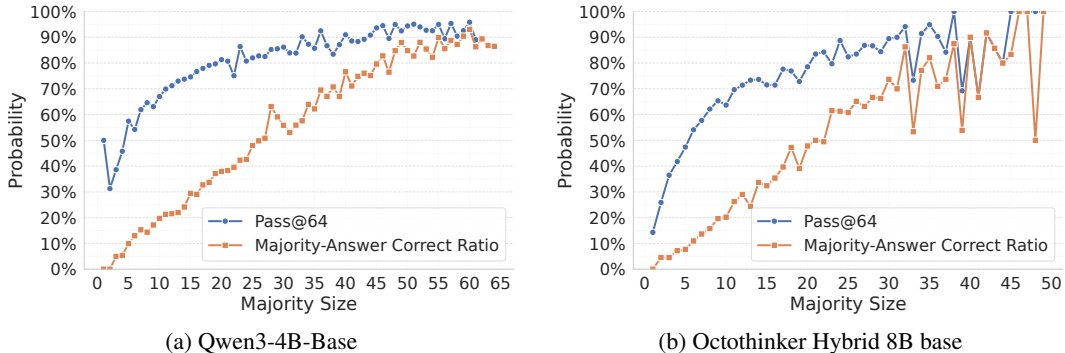

Figure 2: **Majority-Vote Reliability.** Pass@64 and the majority-voted accuracy over 64 samples are compared on the DAPO-MATH dataset for Qwen3-4B-Base (left) and OctoThinker Hybrid-8B-Base (right). The large gap between Pass@64 and majority-vote shows that correct answers often diverge from majority votes. Accuracy also drops sharply when the majority size is small, revealing that majority votes can carry spurious signals. These observations motivate our self-penalizing framework, which seeks robust promising reasoning paths beyond unreliable majority votes.

for the Qwen3-4B-Base model (a) and the OctoThinker Hybrid-8B-Base model (b) on the DAPO-MATH dataset. Although the Majority Correct Ratio rises with the majority vote size (number of solutions that agree), there remains a large gap between Pass@64 and the majority correct ratio, revealing that majority votes can be spurious and often fail to capture the true answer. To bridge this gap, we introduce a pseudo-label weighting scheme. Rather than collapsing all probability mass onto the most frequent answer (majority voting) or distributing it uniformly across candidates, our method assigns weights proportional to the observed vote counts. This produces a consensus distribution that down-weights spurious low-frequency answers while avoiding the brittleness of requiring consensus, providing the foundation for our self-penalization framework.

**Construction**   Given a prompt $x$, we draw $n$ rollouts $\{y_i\}_{i=1}^n \sim \pi_\theta(\cdot \mid x)$ and collect the set of unique answers $\{a_j\}_{j=1}^m$ with counts $c_j$. We treat each $a_j$ as a pseudo label and compute the weighted loss as follows:

$$\mathcal{L}_{\text{GRPO}}(x; \theta) = \sum_{j=1}^m w_j \cdot \mathcal{L}_{\text{GRPO}}(x, a_j; \theta) \tag{2}$$

where $w_j$ is a pseudo-label weight obtained by applying a monotonic function $g$ to frequency $f_j = \frac{c_j}{n}$:

$$w_j = \frac{g(f_j)}{\sum_{\ell=1}^m g(f_\ell)}. \tag{3}$$

We use a Gaussian function centered at the $k \in [0, 1]$ with bias $\sigma > 0$ as our shaping function $g$.

**Interpretation**   Equation 3 prevents collapse to a single majority answer while penalizing spurious low-frequency predictions through a form of *soft selection* over answer frequencies: predictions with higher frequencies receive proportionally larger weights. The skewness of this weighting is controlled by the monotonic shaping function $g(\cdot)$: a steeper $g$ concentrates probability mass on high-frequency answers, whereas a smoother $g$ distributes weight more broadly across answers.

## 2.2   NEGATIVE ROLLOUT PENALIZATION

Existing methods (Zuo et al., 2025; Shafayat et al., 2025) often rely on the majority-voted answer being correct, making low self-consistency regions prone to spurious training signals. Our proposed pseudo-label weighting subsection 2.1 instead leverages control of Pass@n: if any rollout is correct, it provides a valid positive signal, yielding more robust learning under weak consensus. However, when the majority size is very low, Pass@n often degrades because the model may generate no correct rollouts at all. As shown in Figure 2, prompts with very low majority size correspond to unreliable supervision where no answer can be confidently trusted. To handle such cases, we introduce negative rollout penalization, which assumes all responses are incorrect and applies a uniform

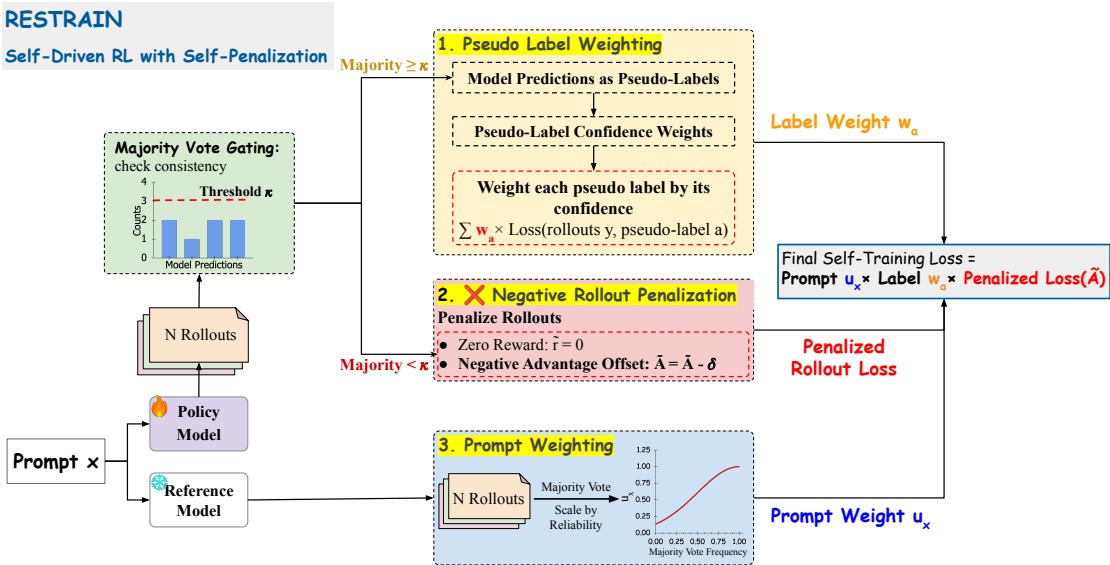

Figure 3: **Overview of Our Method RESTRAIN:** RESTRAIN consists of 3 core components: **1. Pseudo Label Weighting** which takes into account all possible model-predicted answers as candidate pseudo-labels when calculating final losses. **2. Negative Rollout Penalization** which penalizes rollouts with very low confidence by setting zero reward and applying negative advantage offsets to the losses. **3. Prompt Weighting** which downweights entire examples where the reference model predicts with low self-consistency.

negative offset. This reduces explicitly penalizing all rollouts and encourages the model to explore alternative reasoning paths.

**Construction** Consider the GRPO loss term $\mathcal{L}_{\text{GRPO}}(x, a_j; \theta)$ associated with pseudo-label $a_j$. For each rollout $y_i$, denote by $r_{i,j} = R(y_i, a_j)$ the reward and $A_{i,j}$ the corresponding advantage. Let $M(x) = \max_j c_j$ denote the majority count of prompt $x$, where $c_j$ is the vote count for label $a_j$.

When the self-consistency is low ($M(x) < \kappa$), we treat all candidate answers as unreliable, zero out their rewards, and apply a uniform penalty $\delta \geq 0$ to the advantages of all rollouts.

$$\tilde{r}_{i,j} = \begin{cases} r_{i,j} & \text{if } M(x) \geq \kappa \\ 0 & \text{if } M(x) < \kappa \end{cases}, \qquad \tilde{A}_{i,j} = \begin{cases} A_{i,j} & \text{if } M(x) \geq \kappa \\ A_{i,j} - \delta & \text{if } M(x) < \kappa \end{cases} \qquad (4)$$

In PPO/GRPO objectives, this means that all model predictions with $M(x) < \kappa$ contribute only negative updates, penalizing all rollouts with low self-consistency. This discourages reinforcement of spurious majority votes and steers the model away from unreliable reasoning paths.

## 2.3 PROMPT-LEVEL WEIGHTING

Previous penalizing schemes operate at a rollout level. In addition, we introduce a prompt-level penalty. For some prompts, the model exhibits high uncertainty, while for others it produces highly consistent responses. To account for this variation, we scale the update for each prompt by a *fixed* weight that reflects the model's confidence: low-confidence prompts receive smaller updates, and high-confidence prompts receive larger updates. To prevent spurious feedback loops (e.g., inflated confidence during training), these weights are computed once using a frozen base model and kept constant thereafter.

**Construction** For each prompt $x$, we sample $n$ rollouts from the reference policy $\pi_{\text{ref}}$ and compute the majority count $c_{\text{ref}}$. We define the prompt weight again using the monotonic function $g(\cdot)$:

$$u_x = g\left(\frac{c_{\text{ref}}}{n}\right) \qquad (5)$$

We apply $u_x$ to each prompt for all training updates. Unlike pseudo-label weights, prompt-level weights are precomputed offline and remain fixed during the RL training. In Appendix E, we will

Table 1: **On DAPO-14k-Math: RESTRAIN outperforms all unsupervised baselines.** All Pass@1 results(%) are averaged over 16 seeds. The best results are highlighted in **bold**. RESTRAIN outperforms existing baselines without access to gold labels for both Qwen3-4-Base and Octothinker Hybrid-8B Base. In particular, Qwen4-B-Base trained without access to gold labels using RESTRAIN nearly matches the performance of GRPO with gold labels.

| Model | math. | aime25 | olym. | minerva. | mmlu. | gpqa-d. | Avg. ↑ |
|---|---|---|---|---|---|---|---|
| Qwen3-4B-Base | 68.0 | 7.9 | 35.4 | 26.0 | 58.3 | 32.2 | 38.0 |
| *w/ access to gold label* | | | | | | | |
| GRPO | 85.0 | 20.8 | 50.1 | 40.1 | 73.7 | 38.7 | 51.4 |
| *w/o access to gold label* | | | | | | | |
| TTRL | 76.3 | 8.3 | 39.6 | 35.9 | 59.4 | 33.6 | 42.2 |
| SRT (easy prompt) | 77.8 | 7.9 | 39.7 | 36.3 | 60.5 | 34.9 | 42.8 |
| SRT (offline majority label) | 76.9 | 12.0 | 39.8 | 34.2 | 59.4 | 34.5 | 43.1 |
| RESTRAIN (Ours) | **83.0** | **17.9** | **47.0** | **36.5** | **80.9** | **40.2** | **51.0** |
| Δ(RESTRAIN - TTRL) | +6.7 | +9.6 | +7.4 | +0.6 | +21.5 | +6.6 | +8.8 |
| | ↑8.8% | ↑115.7% | ↑18.7% | ↑1.7% | ↑36.2% | ↑19.6% | ↑20.9% |
| OctoThinker Hybrid-8B-Base | 29.8 | 0.8 | 12.1 | 9.3 | 8.6 | 24.6 | 19.2 |
| *w/ access to gold label* | | | | | | | |
| GRPO | 71.7 | 6.2 | 35.2 | 31.3 | 62.0 | 31.0 | 39.6 |
| *w/o access to gold label* | | | | | | | |
| TTRL | 56.5 | 2.7 | 23.2 | 22.1 | 51.7 | 27.3 | 30.6 |
| SRT (offline majority label) | 58.5 | 1.7 | 23.6 | 27.6 | 56.4 | 29.3 | 32.8 |
| RESTRAIN | **62.1** | **6.5** | **24.0** | 26.1 | **65.4** | **30.8** | **35.8** |
| Δ(RESTRAIN - TTRL) | +5.1 | +3.8 | +0.8 | +4.0 | +13.7 | +3.5 | +5.2 |
| | ↑9.0% | ↑140.7% | ↑3.4% | ↑18.1% | ↑26.5% | ↑12.8% | ↑17.0% |

show offline-computed prompt-level weights outperform online variants that are dynamically updated during training.

**Final RESTRAIN loss**    Jointly applying pseudo-label weights $w_j$ from Equation 3 and negative rollout penalization $\tilde{A}_{ij}$ from Equation 4, and the prompt-level weight $u_x$ from Equation 5, we derive our final RESTRAIN loss:

$$\mathcal{L}_{\text{RESTRAIN}}(x;\theta) = u_x \sum_{j=1}^{m} w_j \, \tilde{L}_{GRPO}(x, a_j; \theta) \qquad (6)$$
$$\Downarrow \text{ expand}$$
$$\tilde{L}_{GRPO}(x, a_j; \theta) = -\frac{1}{n} \sum_{i=1}^{n} \min\Big( \rho_i(\theta)\, \tilde{A}_{i,j}, \; \text{clip}\big(\rho_i(\theta), 1-\epsilon, 1+\epsilon\big)\, \tilde{A}_{i,j} \Big)$$
$$- \beta \, \mathbb{D}_{\text{KL}}[\pi_\theta \,\|\, \pi_{\text{ref}}] \qquad (7)$$

## 3    EXPERIMENTAL SETUP

**Datasets**    We evaluate the effectiveness of RESTRAIN on two mathematical and reasoning tasks:

- **DAPO-14k-Math**: We adopt the processed DAPO derived from DAPO-Math-17k (Yu et al., 2025b) which deduplicates prompts and standardizes the formatting of both prompts and reference answers. From this release, we further exclude 3k Chinese language prompts and use 14k English language prompts as our training split, with no further modifications.

- **Synthetic S1k**: A 5k synthetic reasoning dataset from CoT-Self-Instruct (Yu et al., 2025a). Starting from the curated S1k seed set (Muennighoff et al., 2025), Yu et al. (2025a) prompt LLMs to reason step by step and then synthesize new instructions of similar difficulty. Each synthetic example contains both a novel question and a verifiable target answer produced generated by LLM. This dataset complements existing curated math datasets by providing a fully synthetic yet diverse set of reasoning problems, and allows us to systematically test our method under a purely synthetic data generation setting.

**Base Models**    To evaluate the generalizability of our method across different backbone models, we conduct experiments using the following models of various model families and sizes: we use Qwen3-4B-Base and Octothinker Hybrid 8B base (Wang et al., 2025b), which is a specialized, highly optimized reasoning model midtrained from Llama3.1-8B (Dubey et al., 2024), as well as the Llama3.1-8B-Instruct model. More details of experimental settings can be found in Appendix D.

**Benchmarks**    Our benchmark suite comprises six publicly available benchmarks spanning mathematics (four) and science (two). (1) MATH-500 (Hendrycks et al., 2021), (2) AIME25 (Li et al., 2024), (3) OlympiadBench (math subset) (Yang et al., 2024), we use the mathematics portion only. (4) Minerva_math (Yang et al., 2024): the mathematics split from the Minerva quantitative-reasoning suite. (5) MMLU_STEM (Yang et al., 2024), (6) GPQA-Diamond (Yang et al., 2024).

**Metrics**    We evaluate with averaged Pass@1 (Chen et al., 2021) across six benchmarks, sampling 16 predictions per question using a temperature of 0.6 and a top-$p$ value of 0.95 and averaging their 16 Pass@1 accuracies. We use the official evaluation codebase of Qwen2.5-math (Yang et al., 2024).

**Baselines**    We compare RESTRAIN against three recent label-free RLVR methods:

- *TTRL* (Zuo et al., 2025): treats the majority-voted answer as the single pseudo-label, reinforcing it during RL updates. This makes training heavily dependent on the majority being correct, and thus vulnerable to spurious votes.
- *Self-Rewarded Training (SRT)* (Shafayat et al., 2025) proposes two heuristics to mitigate majority-vote collapse:
  - Offline majority label: computes majority votes offline, reducing—but not eliminating—the risk of rewarding self-consistency instead of correctness.
  - Easy prompts: filters training to "easy" prompts with high vote ratios, discarding low-consensus prompts that often contain valuable but underrepresented reasoning paths.
- *Entropy-based Test-Time Reinforcement Learning (ETTRL)* (Liu et al., 2025a) is an entropy-based strategy that improves test-time reinforcement learning for LLM reasoning. We include ETTRL as a baseline only in our Test-Time RL experiments. As the original paper reports results only for test-time training (TTT) and no public implementation is available, we do not extend ETTRL to large-scale label-free RL training (e.g., DAPO-MATH or synthetic S1k).

## 4 MAIN RESULTS

**RESTRAIN outperforms unsupervised baselines**    On DAPO-MATH-14k (Table 1), RESTRAIN - training without gold labels - substantially outperforms existing unsupervised baselines TTRL and SRT. It achieves 51.0%, compared to TTRL (42.2%, +8.8 pp), Offline Majority Label (43.1%, +7.9 pp), and Easy Prompts (42.8%, +8.2 pp). A consistent trend appears on the 5k synthetic corpus (Table 2), where RESTRAIN remains the strongest label-free approach, exceeding the next-best baseline by at least 7.7 pp on average. Notably, when excluding the two science-heavy benchmarks (MMLU_STEM and GPQA-Diamond), RESTRAIN nearly closes the gap with distilling the supervised "reference target" by Qwen3-4B instruct : 45.9% vs. 47.7%, a margin of only 1.8 pp. On OctoThinker Hybrid-8B, we observe the same effect: RESTRAIN consistently surpasses unsupervised baselines TTRL and SRT by large margins. These results underscore the power of self-driven RL with self-penalization, showing that label- and prompt-level penalties transform noisy unlabeled training into signals strong enough to rival gold-label supervision.

**RESTRAIN almost reaches the gold-label upper bound on Qwen3-4B-Base**    In Table 1, we treat the *Gold-label* setting as an empirical upper bound for label-free RLVR, achieving an average accuracy of 51.4%. Remarkably, RESTRAIN reaches 51.0%, trailing by only 0.4 pp—essentially matching supervised GRPO without using labels. Even more striking, RESTRAIN surpasses the gold-label

Table 2: **Synthetic S1k dataset: Our RESTRAIN outperforms all unsupervised baselines.** All Pass@1 results(%) are averaged over 16 seeds. The best results are highlighted in **bold**. When training from Qwen3-4B-Base model on synthetic reasoning tasks without gold label, our method RESTRAIN also outperforms existing unsupervised baselines by 18%.

| Model | math. | aime25 | olym. | minerva. | mmlu. | gpqa-d. | Avg. ↑ |
|---|---|---|---|---|---|---|---|
| Qwen3-4B-Base | 68.0 | 7.9 | 35.4 | 26.0 | 58.3 | 32.2 | 38.0 |
| *w/ access to Qwen3-4B label* | | | | | | | |
| GRPO | 83.7 | 18.9 | 48.4 | 39.7 | 83.6 | 43.5 | 53.0 |
| *w/o access to Qwen3-4B label* | | | | | | | |
| TTRL | 76.0 | 9.2 | 39.3 | 35.9 | 57.6 | 32.8 | 41.8 |
| SRT (easy prompt) | 76.4 | 8.1 | 39.6 | 34.8 | 57.5 | 33.0 | 41.6 |
| SRT (offline majority label) | 75.8 | 10.4 | 39.2 | 33.1 | 57.1 | 33.1 | 41.4 |
| RESTRAIN (Ours) | **81.7** | **20.0** | **45.5** | **36.5** | **73.4** | **40.0** | **49.5** |
| Δ(RESTRAIN - TTRL) | +5.7 | +10.8 | +6.2 | +0.6 | +15.8 | +7.2 | +7.7 |
| | ↑7.5% | ↑117.4% | ↑15.8% | ↑1.7% | ↑27.4% | ↑22.0% | ↑18.4% |

GRPO on MMLU_STEM, scoring **80.9%** vs. 73.7% and on GPQA-Diamond, **40.2%** vs. 38.7%. This suggests strong cross-domain generalization without gold-labels despite being trained solely on the math-focused DAPO-14k dataset. We hypothesize that gold-label supervision encourages overfitting to domain-specific patterns, limiting transfer to science tasks, while RESTRAIN—through self-penalization—relies on distributional signals rather than gold answers, reducing overfitting and preserving generalization across domains.

**RESTRAIN outperforms other Test Time RL Training methods** Test Time RL training focuses on the adaptation to test-time data. We compare our method with recent test-time RL methods like TTRL (Zuo et al., 2025) and Entropy-fork Tree Majority Rollout (ETMR) (Liu et al., 2025a) on LLama3.1-8B-Instruct model, following the same setup as in Liu et al. (2025a), with all methods trained on test prompts without access to gold labels. In Table 3, our approach achieves consistent improvements across challenging math reasoning benchmarks. It surpasses TTRL and ETMR on AMC23 and MATH-

Table 3: **Comparing RESTRAIN v.s. Two Test Time RL Training Methods: TTRL and ETMR on Llama3.1-8B-Instruct.** All results(%) are by greedy decoding following Liu et al. (2025a). RESTRAIN also outperforms the existing test-time scaling method by 11%.

| Test-Time Method | aime24. | amc | math. | Avg. ↑ |
|---|---|---|---|---|
| TTRL | 10.0 | 32.3 | 63.7 | 35.3 |
| ETMR (Liu et al., 2025a) | **16.9** | 35.4 | 59.5 | 37.3 |
| RESTRAIN (Ours) | 16.7 | **40.0** | **67.4** | **41.4** |
| Δ(RESTRAIN - ETMR) | −0.2 | +4.6 | +7.9 | +4.1 |

500 by margins of +13.0% and +13.3%, respectively, yielding an overall +11.0% gain in average accuracy. These results demonstrate that our method can also scale very effectively at test time.

**RESTRAIN can effectively prevent model collapse** Figure 4 shows the averaged Pass@1 on MATH500 across multiple unsupervised methods. The base model is Qwen3-4B-Base, and all methods are trained on the 14k DAPO dataset. We observe that TTRL improves at first but quickly collapses after 50 steps. In contrast, our method RESTRAIN prevents this sudden collapse and keeps training stable throughout. We attribute this stability to RESTRAIN, which does not exclusively reward the majority-vote answer; instead, it assigns soft weights to all distinct answers

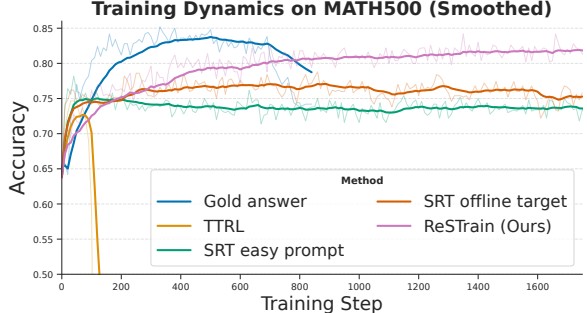

Figure 4: **RESTRAIN has more stable training dynamics.** In contrast to TTRL, our method RESTRAIN steadily improves model performances.

in proportion to their empirical frequencies. This frequency-aware weighting smooths the learning signal, curbs overconfident updates, and mitigates sudden collapse.

# 5 ABLATION STUDY

**Effectiveness of each component in our RESTRAIN**    Table 4 presents the impact of each component of our proposed RESTRAIN. The removal of pseudo-label weighting results in the most substantial performance degradation because training collapses quickly. Omitting negative rollout penalization also hurts performance, reducing the average score from 51.0 to 42.1. Finally, removing prompt-level weighting leads to a more modest performance decrease, yet still validates its positive contribution to the model. Taken together, these results show that all components are necessary for stable and effective unsupervised training.

Table 4: **Each component in RESTRAIN is important.** Each row represents the model's performance with one component removed. The best results are highlighted in bold.

| Model | math. | aime25 | olym. | minerva. | mmlu. | gpqa-d. | Avg. ↑ |
|---|---|---|---|---|---|---|---|
| RESTRAIN | **83.0** | 17.9 | **47.0** | 36.5 | **80.9** | **40.2** | **51.0** |
| (-) Pseudo-label weighting | 67.3 | 6.0 | 34.1 | 24.5 | 59.3 | 33.7 | 37.5 |
| (-) Negative Rollout Penalization | 77.3 | 9.6 | 39.9 | 36.2 | 56.4 | 33.0 | 42.1 |
| (-) Prompt-level weighting | 82.7 | **18.1** | 46.7 | **37.8** | 63.8 | 37.0 | 47.7 |

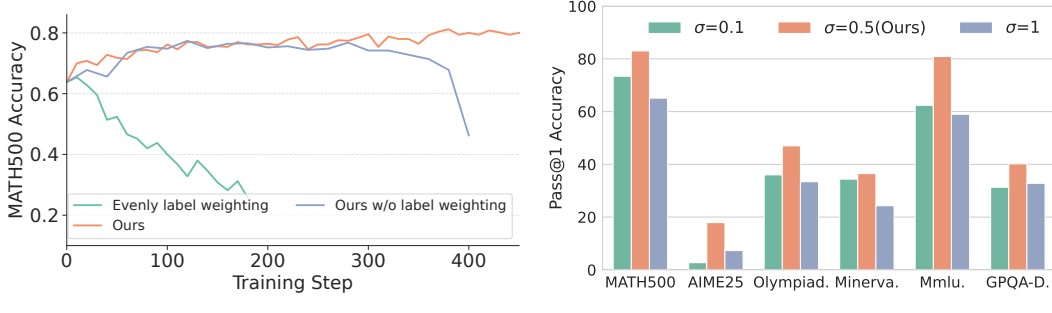

(a) Accuracy curve on MATH500 benchmark.   (b) Performance for different $\sigma$ values.

Figure 5: **Effect of Pseudo-Label Weighting.** Pseudo-label Weighting prevents training collapse, and the hyperparameter $\sigma$ can control the "skewness" of the pseudo-label weight distribution.

**Pseudo-label weighting is crucial to avoid training collapse**    To assess the impact of our pseudo-label weighting module on training and performance, we run two ablation experiments. In the first experiment, we apply prompt-level weighting, negative rollout penalization, and use the majority vote answer as a pseudo label. In the second experiment, we replace the frequency-based soft weights with uniform weights over all targets for each prompt. Figure 5a reports the outcome: without pseudo-label weighting, training becomes unstable and eventually fails. Uniform weighting performs even worse, accelerating degradation and leading to an earlier collapse. This shows that merely considering all targets is insufficient—low-frequency pseudo-labels are typically erroneous/noisy, and assigning them the same weight as high-frequency (likely correct) pseudo-labels can steer the model in the wrong direction. In contrast, frequency-based soft weighting suppresses rare noise and stabilizes training.

**Hyperparameter $\sigma$ in Pseudo-label Weighting**    $\sigma$ controls the "skewness" or concentration of the prompt-level weight distribution. When $\sigma$ is very small, the weighting approaches a step-like function that sharply distinguishes majority from minority answers, effectively behaving like hard majority voting and largely ignoring less frequent responses. In contrast, a large $\sigma$ produces a broad, flat distribution, leading to softer, more evenly spread weights across answers. From Figure 5b, a smaller $\sigma$ ($\sigma = 0.1$) underperforms because it gives too much influence to noisy, infrequent

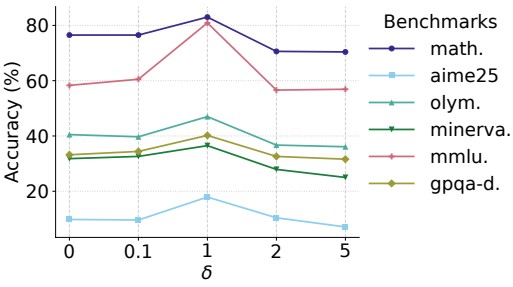 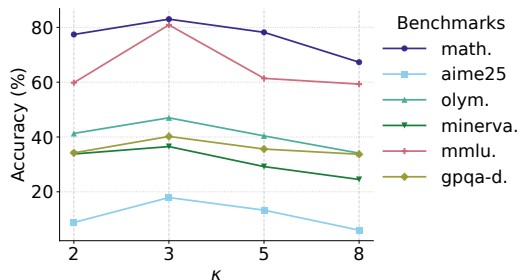

(a) Performance of our RESTRAIN with different negative advantage offset $\delta$.

(b) Performance of our RESTRAIN with different majority count threshold $\kappa$, the number of rollout in our experiment is 16.

Figure 6: **Effect of Pseudo-Label Weighting.** Model performance is sensitive to hyperparameters in Negative Rollout Penalization.

answers. Conversely, a larger $\sigma$ ($\sigma = 1$) is also suboptimal as it fails to leverage valuable signals from correct minority responses. Thus, $\sigma = 0.5$ provides the best balance, effectively filtering noise while retaining the full distributional signal from the model's outputs.

**Hyperparameters in Negative Rollout Penalization** Figure 6a ablates the negative advantage offset $\delta$, which dictates the magnitude of the penalty applied to low-consensus rollouts. The results demonstrate that the model's performance is sensitive to $\delta$. With the penalty disabled ($\delta = 0$), the model simply ignores those low-confidence prompts. Performance is similar to the ablation without Negative Rollout Penalization (Table 4), indicating that simply discarding low-confidence prompts does not hinder training. The best accuracy occurs at $\delta = 1$, suggesting that a moderate penalty effectively discourages the model from generating noisy, low-confidence outputs, thereby stabilizing the training signal and enhancing reasoning capabilities. When the penalty magnitude is increased further to $\delta = 2$ and $\delta = 5$, a consistent and sharp decline in accuracy is observed across all benchmarks. This indicates that an excessively large penalty is detrimental, likely because it over-penalizes the model and may suppress potentially correct, albeit low-frequency, reasoning paths.

Figure 6b varies the majority size threshold $\kappa$ for triggering the negative penalty; the penalty is applied if the count of the most frequent answer is less than $\kappa$. The data reveals a similar trend where performance is suboptimal at both low and high values of $\kappa$, peaking at a value of $\kappa = 3$. A threshold that is too lenient ($\kappa = 2$) fails to penalize many noisy, low-confidence training examples, thus limiting performance improvement. Conversely, a threshold that is too strict ($\kappa = 5$ or 8) suppresses potentially valid reasoning paths in outputs with moderate consensus and causes a significant drop in accuracy. Therefore, the threshold value strikes a crucial balance, effectively filtering unreliable training signals without excessively restricting the model's learning process.

## 6 RELATED WORK

**RL with Verifiable Rewards** RL has shown great promise in improving LLMs, as demonstrated by the success of RL from human feedback (RLHF) and from AI feedback (RLAIF), which aligns model responses with human preferences (Lee et al., 2023; Ouyang et al., 2022; Liu et al., 2025b; Yue et al., 2025). More recently, reinforcement learning with verifiable rewards (Gao et al., 2024; Shao et al., 2024; Guo et al., 2025; Yang et al., 2025; Wen et al., 2025; Song et al., 2025; Team et al., 2025; Fatemi et al., 2025; Wang et al., 2025a; Li et al., 2025b) has been developed to further enhance reasoning capabilities in domains such as mathematics and code. Despite its promise, RLVR is largely limited to settings where a verifiable gold label or exhaustive validators exist, and its outcome-based rewards may limit generalization to tasks that are out of distribution.

**Unsupervised Reward Estimation** Accurately capturing reward signals without relying on human labels has been the focus of several recent studies. Early work like STaR (Zelikman et al., 2022) relies on repeated outcome evaluation. Self-Rewarding LMs (Yuan et al., 2024) explores using LLM-as-a-Judge to provide its own rewards to do self-training. SCPO (Prasad et al., 2024)

introduced self-consistency as an alternative to human-annotated rewards, demonstrating its effectiveness in improving reasoning tasks through (iterative) DPO training. Building on these ideas, TTRL (Zuo et al., 2025) further explored self-consistency signals in an online setting, which treats the majority-voted answer as a pseudo label and leverages the GRPO algorithm (Shao et al., 2024) to update the model. However, TTRL was found to suffer from overconfidence issues, resulting in mode collapse. To address this, SRT (Shafayat et al., 2025) proposed using offline-generated labels and curriculum learning; ETTRL (Liu et al., 2025a) proposed an entropy-based mechanism that enhances the balance between exploration and exploitation, thus mitigating overconfidence and improving overall performance; EVOL-RL (Zhou et al., 2025) introduced novelty reward to increase exploration. Other unsupervised methods derive intrinsic rewards from a model's internal feedback—Reinforcement Learning from Internal Feedback (RLIF). For example, some approaches measure the model's output certainty, using metrics like token- and trajectory-level entropy (Prabhudesai et al., 2025; Agarwal et al., 2025) or self-confidence (Li et al., 2025a). Along these lines, Intuitor (Zhao et al., 2025) utilizes a model's internal confidence termed "self-certainty" as its sole intrinsic reward. Another method, EMPO (Zhang et al., 2025a), uses clustering to extract semantic entropy across multiple rollouts and compute corresponding advantages. Zhang et al. (2025b) theoretically analyzes internal equivalence among RLIF methods and claims that the prior of the base model causes training collapse.

**Unlikelihood Penalization**   Unlikelihood training is a widely adopted technique in neural text generation to penalize undesirable outputs. (Welleck et al., 2019) reduces the probability of specific "negative candidate" tokens. (Li et al., 2019) later employed this approach to improve logical consistency, demonstrating its effectiveness as a general framework for mitigating known biases in dialogue by penalizing a carefully selected set of negative tokens at each generation step. More recently, NSR (Zhu et al., 2025) extended this principle from the neural text generation model to LLMs post-training with their Negative Sampling Rejection (NSR) method. In the context of RLVR, they show that penalizing entire negative trajectories consistently improves performance, preserves generation diversity, and promotes generalization over the base model.

## 7    CONCLUSION

In this paper, we propose RESTRAIN, a self-penalizing reinforcement learning framework that transforms the absence of gold labels into a learning signal, enabling models to self-improve without gold labels. By (i) weighting all predicted targets rather than only the majority, (ii) penalizing low-confidence rollouts within the policy objective, and (iii) discounting prompts with low self-consistency, RESTRAIN enables robust self-improvement and mitigates the training collapse of majority-vote heuristics. Empirically, it delivers more stable optimization and stronger generalization on challenging reasoning tasks like math and science.

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

# A  A PSEUDO CODE OF THE RESTRAIN LOSS FUNCTION

This section shows a pseudo code of our RESTRAIN loss function calculation for one single prompt.

Listing 1: The pseudo-code of the RESTRAIN loss function for one prompt

```python
def restrain_loss(outputs, prompt_weight, threshold, neg_offset):
    # --- Extract answers ---
    answers = [extract_answer(output) for output in outputs]

    # --- Majority size M(x) ---
    counts = Counter(answers)
    Mx = counts.most_common(1)[0][1]

    # --------------------------------
    # Branch 1: Negative penalization
    # --------------------------------
    if Mx < threshold:
        rewards = [0.0] * len(outputs)
        adv = calculate_advantages(rewards)
        adv = [a - neg_offset for a in adv]
        loss = calculate_loss(adv)
        return prompt_weight * loss

    # ---------------------------------
    # Branch 2: Pseudo-label weighting
    # ---------------------------------
    # Calculate label weights
    freqs = counts.values() / len(outputs)
    label_weights = calculate_label_weight(freqs)

    # Calculate each label loss, then weighted sum to a final loss
    final_loss = 0.0
    for i, label in enumerate(counts.keys()):
        rewards = [reward_fn(ans, label) for ans in answers]
        adv = calculate_advantages(rewards)
        loss = calculate_loss(adv)
        final_loss += label_weights[i] * loss

    return prompt_weight * final_loss
```

# B  An Algorithm of the Per-prompt RESTRAIN Loss Function

---

**Algorithm 1:** Per-prompt RESTRAIN Loss

---

**Input**      : Responses $\mathcal{O} = \{o_1, \ldots, o_n\}$; prompt weight $u_x > 0$; majority threshold $\kappa$;
negative offset $\delta \geq 0$.

**Output**    : Loss $L$.

1  $\mathcal{A} = \{a_1, \ldots, a_m\} \leftarrow \texttt{Set}([\texttt{ExtractAnswer}(o_i)]_{i=1}^n); \quad M(x) \leftarrow \max(\texttt{Count}(a))$

2  **if** $M(x) < \kappa$ **then**

3     $r_i \leftarrow 0 \ \forall i; \quad \mathbf{adv} \leftarrow \texttt{CalculateAdvantages}(\{r_i\}_{i=1}^n); \quad adv_i \leftarrow adv_i - \delta \ \forall i;$

4     **return** $L \leftarrow u_x \cdot \texttt{CalculateLoss}(\mathbf{adv})$

5  **else**

6     **for** $j = 1$ **to** $m$ **do**

7        $f_j \leftarrow c(t_j)/n; \quad \tilde{w}_j \leftarrow \texttt{CalculateWeight}(f_j)$

8     $Z \leftarrow \sum_{j=1}^m \tilde{w}_j; \quad w_j \leftarrow \tilde{w}_j/Z \ \forall j;$

9     $L_{\text{final}} \leftarrow 0;$

10     **for** $j = 1$ **to** $m$ **do**

11        $r_i \leftarrow \texttt{RewardFn}(\mathcal{A}[i], a_j) \ \forall i; \quad \mathbf{adv} \leftarrow \texttt{CalculateAdvantages}(\{r_i\}_{i=1}^n);$
      $\ell_j \leftarrow \texttt{CalculateLoss}(\mathbf{adv});$

12        $L_{\text{final}} \leftarrow L_{\text{final}} + w_j \cdot \ell_j$

13     **return** $L \leftarrow u_x \cdot L_{\text{final}}$

---

# C  DISCUSSION OF MOTIVATION

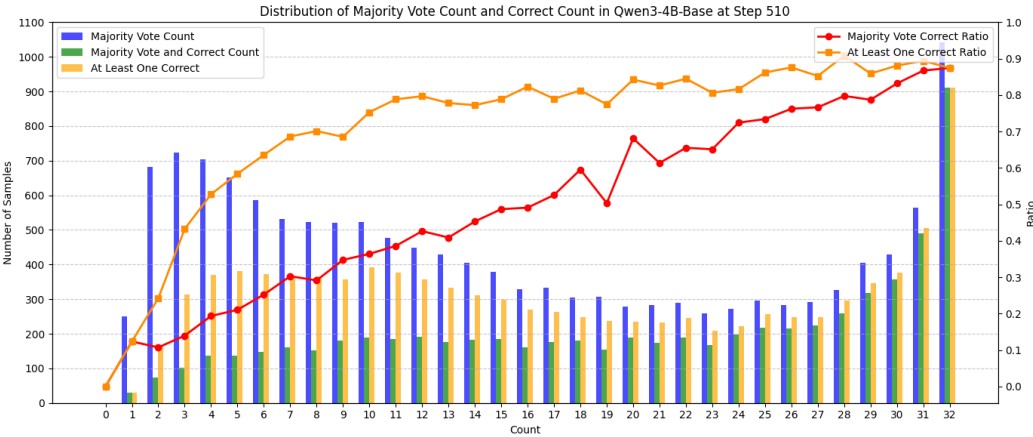

Figure 7: Statistics of Majority Vote Count and Pass@32. The model is trained with Pseudo-label Weighting and Prompt-level weighting. We select a checkpoint when the training converges and use the checkpoint to do inference on the training set to analyze the majority vote count and pass@32.

Figure 7 summarizes majority-vote statistics at step 510 for Qwen3-4B-Base trained with our pseudo-label and prompt-level weighting on the DAPO dataset. The x-axis represents the majority vote count. The chart highlights two key trends: (a) The red line shows the **Majority Vote Correct Ratio**. As the majority vote count decreases (moving left on the graph), the probability that the most frequent answer is actually correct drops almost linearly. (b) The orange line shows the **At Least One Correct Ratio** (i.e. Pass@k). This is the probability that at least one of the generated responses was correct, even if it wasn't the majority answer. This distinction is important for understanding different training methods. A method like TTRL is highly dependent on the majority vote being correct (the red line). When the consensus is low (a low majority vote count), TTRL receives an unreliable and often incorrect training signal. Our proposed method, however, relies on the principle of at least one correct answer being present (the orange line). As long as one of the generated responses is correct, our model receives a valid positive signal for training. This makes it more robust, especially in cases where there isn't a strong consensus on the correct answer. However, the chart also reveals a critical weakness. For very low majority vote counts, the orange line shows a dramatic drop. This indicates that when the model's consensus is extremely low, it's highly probable that none of the generated responses are correct. In this scenario, our method is exposed to significant training noise because there is no positive signal to learn from. To address this specific problem, we introduce our negative rollout penalization to discourage the model from generating sets of answers where none are correct.

# D DETAILED RESULTS

## D.1 BENCHMARKS

Our benchmark suite comprises six publicly available datasets spanning mathematics (four) and science (two). (1) MATH-500(Hendrycks et al., 2021): a 500-problem subset of the MATH corpus, emphasizing competition-style problems across algebra, geometry, number theory, and combinatorics. (2) AIME25 (Li et al., 2024): the official 2025 American Invitational Mathematics Examination questions. (3) OlympiadBench (math subset) (Yang et al., 2024): olympiad-level problems sourced from national/international contests; we use the mathematics portion only. (4) Minerva_math (Yang et al., 2024): the mathematics split from the Minerva quantitative-reasoning suite. (5) MMLU_STEM (Yang et al., 2024): the STEM categories of MMLU (e.g., physics, chemistry, biology, mathematics-adjacent subjects). (6) GPQA-Diamond (Yang et al., 2024): the highest-difficulty split of GPQA with expert-written, graduate-level science questions spanning physics, chemistry, and biology.

In addition to the 6 benchmarks reported in the main paper, we evaluate on three additional benchmarks. They are (1) **AMC23**(Li et al., 2024): prompts drawn from the 2023 American Mathematics Competitions, covering core high-school problem-solving domains. (2) **AIME24** (Li et al., 2024): the official 2024 American Invitational Mathematics Examination questions. (3) **s1k (verifiable subset)** (Muennighoff et al., 2025): a subset of 893 s1k examples with verifiable answers from Yu et al. (2025a).

## D.2 IMPLEMENTATION DETAILS

We implement TTRL, SRT, and RESTRAIN using the VERL codebase. To validate correctness, we reproduce a representative experiment from the original papers with our implementations and verify that the resulting accuracies match. Since ETMR has not released code, we report its results as stated in the original paper. For hyperparameters, we use a learning rate of $1 \times 10^{-6}$, and adopt the AdamW optimizer for the policy model. We set kl loss coefficient to 0.001, and the entropy coefficient to 0. For rollout, we sample 16 responses using a temperature of 1.0 for training. The maximum generation length is set to 4096 for Qwen3-4B-Base and Llama3.1-8B-Instruct, and 8192 for Octothinker Hybrid 8B base model. We employ a unified hyperparameter configuration for RESTRAIN across all experiments. Specifically, we set the mean for the pseudo-label/prompt weight to 1.0, the bias $\sigma = 0.5$, the negative advantage offset $\delta = 1.0$, and the majority size threshold $\kappa = 3$. We set the number of epochs to 20. All experiments were conducted on 32 * NVIDIA A100 80GB GPUs.

## D.3 ADDITION RESULTS

Table 5 and 6 show full results of our RESTRAIN on nine benchmarks. Results show that our method can outperform all unsupervised methods on both Qwen3-4B-Base and Octothinker Hybrid 8B base models with two different training datasets.

Table 7 show experimental results on three different capacity models: a base model: Qwen3-1.7B-Base, a math-specific model: Qwen2.5-math-7B, and an instruct model: Llama-3.1-8B-Instruct, cross two training datasets(DAPO-14k-math and NuminaMath-10k). Consistent with our findings in the main result section, RESTRAIN outperforms the TTRL baseline under all settings. This confirms that our self-penalization mechanism is also effective for instruct models. By validating across Qwen (Base and math), OctoThinker (Specialized Mid-trained), and Llama (Instruct), we demonstrate that RESTRAIN generalizes across model families and training datasets.

Table 5: The table shows the evaluation results of training Qwen3-4B-Base on **14k DAPO dataset**, all results(%) are averaged over 16 seeds. The best results are highlighted in **bold**.

| Model | math. | amc. | aime24 | aime25 | olym. | miner. | mmlu. | gpqa. | s1k | avg |
|---|---|---|---|---|---|---|---|---|---|---|
| Qwen3-4B-Base | 68.0 | 45.6 | 10.4 | 7.9 | 35.4 | 26.0 | 58.3 | 32.2 | 5.1 | 32.1 |
| *w/ access to gold label* | | | | | | | | | | |
| GRPO | 85.0 | 69.3 | 21.2 | 20.8 | 50.1 | 40.1 | 73.7 | 38.7 | 12.2 | 45.7 |
| *w/o access to gold label* | | | | | | | | | | |
| TTRL | 76.3 | 52.6 | 12.0 | 8.3 | 39.6 | 35.9 | 59.4 | 33.6 | 4.6 | 35.8 |
| SRT (easy prompt) | 77.8 | 52.3 | 13.5 | 7.9 | 39.7 | 36.3 | 60.5 | 34.9 | 5.6 | 36.5 |
| SRT (offline majority label) | 76.9 | 51.8 | 10.4 | 12.0 | 39.8 | 34.2 | 59.4 | 34.5 | 4.7 | 36.0 |
| RESTRAIN | **83.0** | **60.2** | **20.3** | **17.9** | **47.0** | **36.5** | **80.9** | **40.2** | **10.3** | **44.0** |
| Oct.Hybrid-8B-Base | 29.8 | 16.1 | 1.9 | 0.8 | 12.1 | 9.3 | 8.6 | 24.6 | 2.1 | 15.0 |
| *w/ access to gold label* | | | | | | | | | | |
| GRPO | 71.7 | 49.4 | 10.8 | 6.2 | 35.2 | 31.3 | 62.0 | 31.0 | 7.2 | 33.9 |
| *w/o access to gold label* | | | | | | | | | | |
| TTRL | 56.5 | 32.2 | 3.9 | 2.7 | 23.2 | 22.1 | 51.7 | 27.3 | 3.5 | 24.8 |
| RESTRAIN | **61.6** | **33.6** | **6.0** | **8.5** | **24.6** | **25.0** | **64.6** | **29.9** | **4.4** | **28.7** |

Table 6: The table shows the evaluation results of training Qwen3-4B-Base on **5k Synthetic S1k dataset**, all results(%) are averaged over 16 seeds. The best results are highlighted in **bold**.

| Model | math. | amc. | aime24 | aime25 | olym. | miner. | mmlu. | gpqa. | s1k | Avg. ↑ |
|---|---|---|---|---|---|---|---|---|---|---|
| Qwen3-4B-Base | 68.0 | 45.6 | 10.4 | 7.9 | 35.4 | 26.0 | 58.3 | 32.2 | 5.1 | 32.1 |
| *w/ access to Qwen3-4B label* | | | | | | | | | | |
| GRPO | 83.7 | 64.5 | 23.7 | 18.9 | 48.4 | 39.7 | 83.5 | 43.5 | 11.5 | 46.4 |
| *w/o access to Qwen3-4B label* | | | | | | | | | | |
| TTRL | 76.0 | 50.2 | 10.8 | 9.2 | 39.3 | 35.9 | 57.6 | 32.8 | 4.8 | 35.2 |
| SRT (easy prompt) | 76.4 | 52.3 | 12.1 | 8.1 | 39.6 | 34.8 | 57.5 | 33.0 | 4.9 | 35.4 |
| SRT (offline majority label) | 75.8 | 53.3 | 11.9 | 10.4 | 39.2 | 33.1 | 57.1 | 33.1 | 4.6 | 35.4 |
| RESTRAIN | **81.7** | **58.4** | **17.9** | **20.0** | **45.5** | **36.5** | **73.4** | **40.0** | **8.8** | **42.5** |

Table 7: Additional results on a base model, a math specific model and an instruct model cross two training datasets.

| Model & Dataset | Method | MATH500 | AMC23 | AIME24 | AIME25 | Olym. | Avg |
|---|---|---|---|---|---|---|---|
| **Qwen2.5-Math-7B** *(10k NuminaMath)* | Base | 52.8 | 44.0 | 16.6 | 3.3 | 17.8 | 26.9 |
| | Gold | 79.7 | 64.0 | 28.0 | 14.6 | 39.6 | 45.1 |
| | TTRL | 77.9 | 61.5 | 18.0 | 8.0 | 37.2 | 40.5 |
| | RESTRAIN | **79.3** | **62.5** | **26.7** | **13.9** | **40.7** | **44.6** |
| **Qwen3-1.7B-Base** *(10k NuminaMath)* | Base | 56.4 | 30.1 | 4.5 | 3.9 | 22.8 | 23.5 |
| | Gold | 69.9 | 42.0 | 9.7 | 3.1 | 31.5 | 31.2 |
| | TTRL | 63.2 | 36.2 | 5.4 | 3.5 | 26.0 | 26.8 |
| | RESTRAIN | **66.9** | **41.0** | **8.3** | **5.2** | **29.6** | **30.2** |
| **Llama-3.1-8B-Inst.** *(14k DAPO Math)* | Base | 49.6 | 24.4 | 5.4 | 0.8 | 17.2 | 19.4 |
| | Gold | 53.4 | 28.9 | 6.4 | 0.2 | 20.2 | 21.8 |
| | TTRL | 47.5 | 25.0 | 6.1 | 0.8 | 17.3 | 19.3 |
| | RESTRAIN | **51.2** | **26.4** | **7.1** | **1.1** | 17.3 | **20.6** |

# E    ABLATION STUDY OF PROMPT-LEVEL WEIGHTING

In this section, we evaluate the effect of prompt-level weighting on training. We ablate it by two experiments, one is comparing our offline prompt weighting with online prompt weighting, another is setting all prompt weights to 1 ("w/o prompt weighting"). As shown in Figure 8, in the first experiment, online prompt weighting quickly collapses while offline prompt weighting can continue to improve. For the second experiment, both methods' accuracies improve quickly at the start. Initially, the model without prompt weighting learns slightly faster. However, our method soon overtakes it and consistently maintains a higher accuracy. Notably, the performance of the model without prompt weighting becomes unstable and drops sharply after 1,500 training steps. In contrast, our method's accuracy remains stable and continues to improve. This suggests that offline prompt-level weighting is key to achieving both higher final accuracy and greater training stability.

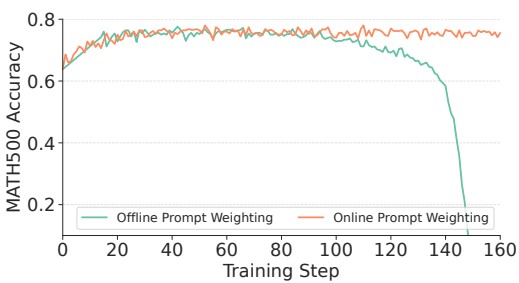 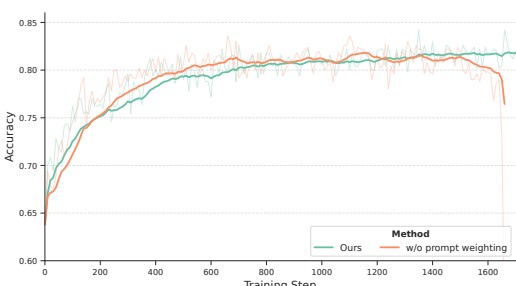

(a) Online Prompt Weighting collapses very quickly at around 100 steps.

(b) Without Prompt Weighting, the model will ultimately collapse.

Figure 8: **Offline Prompt-weighting can help model train stable.**

# F    STUDY OF HYPERPARAMETER WEIGHT BIAS $\sigma$ IN PSEUDO-LABEL WEIGHTING

In this section, we examine the bias parameter $\sigma$ used in pseudo-label weighting. Table 8 reports the tuning results. When $\sigma$ is small, the scheme effectively reduces to selecting the majority-vote answer as the pseudo label; when $\sigma$ is large, it approaches uniform weighting. We observe that $\sigma$ values near zero or above 1 lead to training collapse and substantially worse performance, whereas a moderate setting (e.g., $\sigma = 0.5$) yields the best stability and accuracy.

Table 8: **Ablation of Pseudo-label Weighting.** The table shows the evaluation results of training Qwen3-4B-Base on **14k DAPO-Math dataset** by varying the hyperparameter weight bias, all results(%) are averaged over 16 seeds. The best results are highlighted in **bold**.

| Target Level Weighting Bias $\sigma$ 
 Small $\sigma$ = skewed on majority label 
 Large $\sigma$ = evenly dist. on all labels | math. | aime25 | olym. | minerva. | mmlu. | gpqa-d. | Avg. ↑ |
|---|---|---|---|---|---|---|---|
| $\sigma = 0$ (0 weights on non-majority labels) | 67.8 | 7.7 | 34.7 | 24.1 | 58.6 | 32.1 | 37.5 |
| $\sigma = 0.1$ | 73.4 | 2.7 | 36.0 | 34.4 | 62.4 | 31.3 | 40.0 |
| $\sigma = 0.25$ | 76.5 | 9.6 | 39.7 | 32.6 | 60.52 | 34.4 | 42.2 |
| $\sigma = 0.5$ | **83.0** | **17.9** | **47.0** | **36.5** | **80.9** | **40.2** | **51.0** |
| $\sigma = 1$ | 65.1 | 7.3 | 33.4 | 24.3 | 59.0 | 32.8 | 37.0 |
| $\sigma = 2$ | 66.2 | 6.2 | 33.1 | 23.8 | 58.9 | 31.4 | 36.6 |
| $\sigma = 5$ | 61.1 | 5.8 | 32.6 | 23.7 | 58.4 | 33.3 | 35.8 |
| $\sigma = \infty$ (evenly distributed) | 66.8 | 6.9 | 34.6 | 24.6 | 59.8 | 32.9 | 37.6 |

# G  STUDY OF HYPERPARAMETER NEGATIVE ADVANTAGE OFFSET $\delta$ AND MAJORITY COUNT THRESHOLD $\kappa$

In this section, we examine how the negative-advantage offset $\delta$ and the majority-count threshold $\kappa$ influence performance. The offset $\delta$ scales the penalty applied to low-consensus rollouts; if set too high, it over-penalizes the policy and induces a sharp accuracy decline. The threshold $\kappa$ decides which prompts are treated as low-consensus: a strict threshold discards many informative examples and hurts accuracy, while an overly loose threshold admits noisy cases and weakens the intended penalization. Appropriate, balanced choices of $\delta$ and $\kappa$ suppress noise without sacrificing useful signal.

Table 9: **Results on Different Negative rollout Penalty.** The table shows the evaluation results of training Qwen3-4B-Base on **14k DAPO-Math dataset** by varying the negative advantage offset, all results(%) are averaged over 16 seeds. The best results are highlighted in **bold**.

| Negative Advantage Offset | math. | aime25 | olym. | minerva. | mmlu. | gpqa-d. | Avg. ↑ |
|---|---|---|---|---|---|---|---|
| $\delta = 0$ | 76.5 | 9.8 | 40.5 | 31.8 | 58.3 | 33.2 | 41.7 |
| $\delta = 0.1$ | 78.7 | 13.3 | 40.8 | 36.5 | 59.3 | 35.5 | 44.0 |
| $\delta = 1$ | **83.0** | **17.9** | **47.0** | **36.5** | **80.9** | **40.2** | **51.0** |
| $\delta = 2$ | 70.6 | 10.4 | 36.7 | 27.9 | 56.6 | 32.6 | 39.1 |
| $\delta = 5$ | 70.4 | 7.1 | 36.1 | 25.0 | 56.9 | 31.6 | 37.9 |

Table 10: **Results on Different Majority Count Threshold.** The table shows the evaluation results of training Qwen3-4B-Base on **14k DAPO-Math dataset** by varying the weight bias, all results(%) are averaged over 16 seeds. The best results are highlighted in **bold**.

| Majority Count Threshold (for negative rollouts) | math. | aime25 | olym. | minerva. | mmlu. | gpqa-d. | Avg. ↑ |
|---|---|---|---|---|---|---|---|
| $\kappa = 2$ | 77.4 | 8.8 | 41.3 | 33.8 | 59.8 | 34.2 | 42.5 |
| $\kappa = 3$ | **83.0** | **17.9** | **47.0** | **36.5** | **80.9** | **40.2** | **51.0** |
| $\kappa = 5$ | 78.2 | 13.3 | 40.4 | 29.2 | 61.4 | 35.6 | 43.0 |
| $\kappa = 8$ | 67.3 | 6.0 | 34.1 | 24.5 | 59.3 | 33.7 | 37.5 |

To further discuss $\kappa$, in our experiments, we fixed the rollout number at $n = 16$ and the sampling temperature at 1.0. Crucially, we utilized a fixed $\kappa = 3$ across all different training datasets (DAPO-14k, Synthetic S1k) and models (Qwen3-4B-Base, OctoThinker-8B-Hybrid-Base). This single configuration consistently outperformed baselines, suggesting that $\kappa = 3$ serves as a robust generalist setting within the standard rollout regime ($n = 16$).

However, $\kappa$ is naturally coupled with the rollout number ($n$) and the model's capability. $\kappa$ acts as a gate for Negative Rollout Penalization, defining the minimum consistency required to consider a prompt's signal "reliable" enough to avoid penalization. Relation to Rollout Number ($n$): If $n$ is increased significantly (e.g., from 16 to 100), $\kappa$ should likely be scaled to represent a similar ratio of self-consistency. Relation to Difficulty/Assumption of Negative Trajectories: Increasing $\kappa$ represents a stronger assumption regarding negative trajectories. A higher $\kappa$ treats a larger portion of low-consensus outputs as "noise" to be penalized. For extremely hard tasks where even correct answers rarely achieve consensus, a lower $\kappa$ might be necessary to avoid suppressing the rare correct signal. Conversely, for easier tasks where the model is generally confident, a higher $\kappa$ could further enforce strict consistency. While our results show that $\kappa = 3$ is empirically stable, we acknowledge that $\kappa$ remains a tunable hyperparameter that governs the trade-off between suppressing noise and preserving minority signals.

# H    DISCUSSION OF PASS@1 VS. MAJORITY VOTE PERFORMANCE

In this section, we want to study how the gap between Pass@1 and Majority Voting evolves. We conduct experiments on the OctoThinker-8B-Hybrid-Base model. We compared the Base model, TTRL, and RESTRAIN across four benchmarks.

The results (detailed in Table 11) reveal three distinct critical findings:

- *RESTRAIN bridges the"Consistency Gap":* The Base model exhibits a massive discrepancy between Pass@1 and Majority Vote (e.g., a 34.6% gap on MATH500). This indicates the model often possesses the knowledge in its latent distribution but fails to output it reliably in a single attempt. RESTRAIN drastically reduces this gap (e.g., to 12.7% on MATH500), effectively converting the model's latent "majority potential" into reliable, single-shot performance.

- *RESTRAIN expands the "Knowledge Boundary" (Raising the Ceiling):* A key limitation of TTRL is that it often only aligns the model with its existing majority, yielding minimal gains in the upper bound. On MATH500, TTRL only improved the Majority Vote by 3.0% (64.4% → 67.4%). At the same time, RESTRAIN increased the Majority Vote by 10.4% (64.4% → 74.8%). This demonstrates that RESTRAIN significantly enhances the model's reasoning capabilities, generating correct reasoning paths that were not dominant in the base model.

- *RESTRAIN prevents "Ceiling Collapse" on Hard Tasks:* On the most challenging benchmark, AIME24, we observe a critical failure in TTRL. TTRL caused the Majority Vote to drop below the Base model's performance (Base: 10.0% → TTRL: 6.67%). This suggests TTRL overfitted to spurious signals or easy patterns, degrading the model's ability to solve hard problems. In contrast, RESTRAIN successfully raised the ceiling to 16.67%. This proves our RESTRAIN protects against the model collapse often seen in TTRL on difficult training tasks.

Table 11: **Pass@1 vs. Majority Vote (Maj) Performance.** We calculate the *Gap* as Maj Vote − Pass@1 to highlight consistency.

| Benchmark | Metric | OctoThinker-8B-Hybrid | TTRL | RESTRAIN (Ours) |
|---|---|---|---|---|
| **MATH500** | Pass@1 | 29.8 | 56.5 | **62.1** |
| | Maj Vote | 64.4 | 67.4 | **74.8** |
| | *Gap* | *34.6* | *10.9* | *12.7* |
| **OlympiadBench** | Pass@1 | 12.1 | 23.2 | **24.0** |
| | Maj Vote | 29.3 | 33.9 | **35.7** |
| | *Gap* | *17.2* | *10.7* | *11.7* |
| **Minerva Math** | Pass@1 | 9.3 | 22.1 | **26.1** |
| | Maj Vote | 25.0 | 35.3 | **37.9** |
| | *Gap* | *15.7* | *13.2* | *11.8* |
| **AIME24** | Pass@1 | 1.88 | 3.94 | **6.46** |
| | Maj Vote | 10.0 | 6.67 | **16.67** |
| | *Gap* | *8.12* | *2.73* | *10.21* |

# I DISCUSSION OF COMPUTATIONAL COST

In this section, we want to discussion the computational cost of RESTRAIN. We claim that RESTRAIN has nearly identical computational overhead to TTRL. Both methods share the same training hyperparameters—specifically rollout number, maximum sequence length, batch size, and number of epochs. Consequently, the most resource-intensive operations (LLM generation and policy forward/backward passes) are identical.

RESTRAIN introduces only negligible overhead through lightweight operations on the generated rollouts: grouping unique answers, computing normalized pseudo-label weights, and applying consensus-gated offsets. These are simple operations that do not require additional model passes or parameters. Furthermore, the prompt-level weights are derived from a one-time offline computation, adding only a small constant setup cost rather than a recurring per-step burden.

To validate this, we conducted a runtime analysis on the Qwen3-4B-Base model, which was trained on the DAPO-14k-MATH dataset (see Table 12). The results confirm that RESTRAIN maintains a training time per step comparable to both TTRL and standard gold-label training. While TTRL may exhibit shorter total training time due to early stopping caused by model collapse, the computational cost per step remains equivalent.

Table 12: **Computational Cost Analysis.** Comparison of training time per step and average response length.

| Method | Training Time (Step 1) | Avg. Response Length |
|---|---|---|
| Train with Gold Label | 144s | 758.0 |
| TTRL | 216s | 746.8 |
| **RESTRAIN (Ours)** | **155s** | **776.8** |

## J    DISCUSSION OF ADAPTING RESTRAIN TO PPO

The RESTRAIN framework is inherently designed for group-based reinforcement learning methods like GRPO. Since these methods already compute baselines relative to a group of rollouts, integrating RESTRAIN's soft-weighting and penalization logic is seamless. However, adapting this framework to value-based RL methods like PPO requires more intricate design choices. Specifically, because PPO relies on a learned Critic ($V_\phi$) rather than group averages for variance reduction, we must explicitly define how to train this Critic to interpret RESTRAIN's signals without destabilizing the advantage estimation. We show a potential adaption below as an example.

To apply RESTRAIN to PPO, we must translate its group-level signals into scalar rewards that a learned Critic ($V_\phi$) can predict. The key shift is replacing the standard unsupervised "hard majority" baseline, where the most frequent answer gets a reward of 1 and others 0, with RESTRAIN's "soft consensus" approach. The PPO Critic ($V_\phi$) is tasked with learning the expected consensus score rather than a binary success probability. This allows the advantage function $A(s, a) = r - V(s)$ to correctly capture nuance: a rollout matching a strong consensus yields a positive advantage, while a rollout matching a weak consensus yields a smaller signal. To prevent the model from reinforcing "hallucinated majorities" where the group is confused (i.e., the majority count $M(x)$ is below a threshold $\kappa$), we intervene directly in the advantage estimation to apply a "penalization". Specifically, when the model is confused ($M(x) < \kappa$), we override the standard calculation with a penalty. We zero out the rewards for the group and inject a negative offset $\delta$, resulting in a final advantage calculation:

$$A_{\text{final}} = A_{\text{GAE}}(0) - \delta$$

Crucially, the Critic must be trained on the *unpenalized* rewards rather than the penalty itself. This ensures $V(s) \approx 0$, preserving the pure negative signal $-\delta$ in the policy update. Finally, the entire PPO loss is scaled by an offline prompt-reliability score $u_x$, derived from a reference model, ensuring gradients are only applied on solvable prompts:

$$\mathcal{L} = u_x \cdot \mathcal{L}_{\text{PPO}}(A)$$

## K  THE USE OF LARGE LANGUAGE MODELS

In this work, we use LLM for writing polishing and do not use it for any other purpose.

