# OpenReview forum: "RESTRAIN: From Spurious Votes to Signals — Self-Training RL with Self-Penalization"
_ICLR.cc/2026/Conference — ICLR 2026 Poster_

### Official Review · Reviewer_hpMx · 2025-10-17

**Soundness:** 4
**Presentation:** 3
**Contribution:** 4
**Rating:** 8
**Confidence:** 4

**Summary:**

The paper proposes RESTRAIN, an RLVR framework that does not need ground truth answers for questions. It tackles the problem that majority-voted answer may not be correct, especially for hard labels, and designed a weight-smoothing method to consider infrequent answers (pseudo-label weighting, negative rollout penalization and prompt-level weighting). The experiments show relatively large improvements over other unsupervised baselines.

**Strengths:**

- The motivation is very clear (majority voting != correct) and the designed soft weighting of infrequent answers seem reasonable and directly tackle the problem of majority-voting.
- The negative rollout penalization is also convincing in down-weighting too infrequent/long-tailed answers (where the model is unsure about).
- From the experiment results, it seems the proposed RESTRAIN can largely improve over existing unsupervised RLVR methods like TTRL. Besides, they test on both Qwen3-4B-Base model and Octothinker Hybrid 8B, which demonstrate that the method is applicable to recent SOTA base models and various model sizes.
- They also provided very detailed ablation studies and useful details for reproduction.

**Weaknesses:**

- Not sure if the threshold kappa will be sensitive to rollout numbers and sampling temperature. Is it fixed over different datasets (which have different difficulty)?
- Typo in Table 1: w/ access to gold labe -> w/ access to gold label

**Questions:**

- How does the gap between pass@1 and majority voting change after RESTRAIN training?

---

> ### Author Response · Authors · 2025-11-26
> **Responses to Reviewer hpMx (Part 1)**
>
> We appreciate the reviewer’s insightful comments, which have helped us strengthen our experiments and clarify our contribution. We address each specific question below.
>
> Q1. How threshold kappa may change according to different tasks & models.
>
> We thank the reviewer for this insightful question regarding the relationship between the threshold $\kappa$, rollout numbers, and task difficulty.
> 1. Empirical Robustness (Fixed $\kappa$ across tasks):
> In our experiments, we fixed the rollout number at $n=16$ and the sampling temperature at $1.0$. Crucially, we utilized a fixed $\kappa=3$ across all different training datasets (DAPO-14k, Synthetic S1k) and models (Qwen3-4B-Base, OctoThinker-8B-Hybrid-Base). This single configuration consistently outperformed baselines, suggesting that $\kappa=3$ serves as a robust generalist setting within the standard rollout regime ($n=16$).
> 2. Theoretical Sensitivity & Tuning:
> We agree with the reviewer that $\kappa$ is naturally coupled with the rollout number ($n$) and the model's capability. $\kappa$ acts as a gate for Negative Rollout Penalization, defining the minimum consistency required to consider a prompt's signal "reliable" enough to avoid penalization.
> Relation to Rollout Number ($n$): If $n$ is increased significantly (e.g., from 16 to 100), $\kappa$ should likely be scaled to represent a similar ratio of self-consistency.
> Relation to Difficulty/Assumption of Negative Trajectories: Increasing $\kappa$ represents a stronger assumption regarding negative trajectories. A higher $\kappa$ treats a larger portion of low-consensus outputs as "noise" to be penalized. For extremely hard tasks where even correct answers rarely achieve consensus, a lower $\kappa$ might be necessary to avoid suppressing the rare correct signal. Conversely, for easier tasks where the model is generally confident, a higher $\kappa$ could further enforce strict consistency.
> While our results show that $\kappa=3$ is empirically stable, we acknowledge that $\kappa$ remains a tunable hyperparameter that governs the trade-off between suppressing noise and preserving minority signals.
>
> We will revise the manuscript to add this discussion about $\kappa$.
>
> Q2. Typos
>
> We thank the reviewer for pointing out this typo. We will revise it in our final version of the manuscript.

---

> > ### Author Response · Authors · 2025-11-26
> > **Responses to Reviewer hpMx (Part 2)**
> >
> > Q3. How does the gap between pass@1 and majority voting change after RESTRAIN training?
> >
> > We thank the reviewer for this insightful question. To address how the gap between Pass@1 and Majority Voting evolves, we conducted additional evaluations using the OctoThinker-8B-Hybrid-Base model. We compared the Base model, TTRL, and RESTRAIN across four benchmarks.
> > The results (detailed in Table) reveal three distinct critical findings:
> > 1. RESTRAIN bridges the "Consistency Gap":
> > The Base model exhibits a massive discrepancy between Pass@1 and Majority Vote (e.g., a 34.6% gap on MATH500). This indicates the model often possesses the knowledge in its latent distribution but fails to output it reliably in a single attempt. RESTRAIN drastically reduces this gap (e.g., to 12.7% on MATH500), effectively converting the model's latent "majority potential" into reliable, single-shot performance.
> > 2. RESTRAIN expands the "Knowledge Boundary" (Raising the Ceiling):
> > A key limitation of TTRL is that it often only aligns the model with its existing majority, yielding minimal gains in the upper bound.
> > On MATH500, TTRL only improved the Majority Vote by 3.0% (64.4% $\rightarrow$ 67.4%).
> > At the same time, RESTRAIN increased the Majority Vote by 10.4% (64.4% $\rightarrow$ 74.8%).
> > This demonstrates that RESTRAIN significantly enhances the model's reasoning capabilities, generating correct reasoning paths that were not dominant in the base model.
> > 3. RESTRAIN prevents "Ceiling Collapse" on Hard Tasks:
> > On the most challenging benchmark, AIME24, we observe a critical failure in TTRL.
> > TTRL caused the Majority Vote to drop below the Base model's performance (Base: 10.0% $\rightarrow$ TTRL: 6.67%). This suggests TTRL overfitted to spurious signals or easy patterns, degrading the model's ability to solve hard problems.
> > In contrast, RESTRAIN successfully raised the ceiling to 16.67%. This proves our RESTRAIN protects against the model collapse often seen in TTRL on difficult training tasks.
> >
> > We will include this analysis in the revision to illustrate how RESTRAIN effectively regularizes the model's output distribution, closing the consistency gap while avoiding the performance degradation seen in TTRL on difficult tasks.
> >
> > ### **Table: Pass@1 vs. Majority Vote (Maj) Performance**
> >
> > | **Benchmark** | **Metric** | **OctoThinker-8B-Hybrid** | **TTRL** | **RESTRAIN (Ours)** |
> > | :--- | :--- | :--- | :--- | :--- |
> > | **MATH500** | Pass@1 | 29.8 | 56.5 | **62.1** |
> > | | Maj Vote | 64.4 | 67.4 | **74.8** |
> > | | *Gap* | *34.6* | *10.9* | *12.7* |
> > | **OlympiadBench** | Pass@1 | 12.1 | 23.2 | **24.0** |
> > | | Maj Vote | 29.3 | 33.9 | **35.7** |
> > | | *Gap* | *17.2* | *10.7* | *11.7* |
> > | **Minerva Math** | Pass@1 | 9.3 | 22.1 | **26.1** |
> > | | Maj Vote | 25.0 | 35.3 | **37.9** |
> > | | *Gap* | *15.7* | *13.2* | *11.8* |
> > | **AIME24** | Pass@1 | 1.88 | 3.94 | **6.46** |
> > | | Maj Vote | 10.0 | 6.67 | **16.67** |
> > | | *Gap* | *8.12* | *2.73* | *10.21* |
> >
> > *(Note: We calculated the Gap as `Maj Vote - Pass@1` to highlight the consistency analysis.)*

---

### Official Review · Reviewer_7W3F · 2025-10-31

**Soundness:** 3
**Presentation:** 3
**Contribution:** 3
**Rating:** 6
**Confidence:** 3

**Summary:**

This work aims to solve the training instability and collapse caused by unreliable majority-vote signals in unsupervised RL methods like TTRL. The paper proposes RESTRAIN, a self-penalizing framework that replaces TTRL's reward scheme. Its novel loss function stabilizes training by using soft, frequency-based weighting for all potential answers, actively penalizing low-consistency outputs, and down-weighting unreliable prompts identified by a frozen model. Experiments show RESTRAIN significantly outperforms TTRL, avoids training collapse, and nearly matches the performance of a fully supervised, gold-label baseline.

**Strengths:**

1. The work directly addresses the critical and well-documented stability problem of majority-vote-based unsupervised RL.

2. The three components are well-motivated and logically designed to counteract the specific failure modes of TTRL, such as signal smoothing and noise penalization.

3. The empirical results are a significant strength, demonstrating performance that not only avoids collapse but also approaches the upper bound of fully supervised gold-label training.

4. The paper is supported by a thorough set of ablation studies that demonstrate the necessity of each component for stable training and final performance.

**Weaknesses:**

1. High Hyperparameter Sensitivity: The method's performance appears highly sensitive to its key hyperparameters, including the weighting skewness, the penalty threshold, and the penalty offset. The ablation studies show that performance drops sharply outside a narrow range of these values, which may hinder its general applicability and reproducibility.

2. Limited Novelty: The contribution appears to be more of a successful systems-level engineering effort than a fundamentally new paradigm. The core components (soft weighting, curriculum learning, and negative penalization) are all established concepts. Overall, this work is more like an incremental "patch" on TTRL.

**Questions:**

1. The hyperparameters are clearly critical. How were the optimal values selected?

2. Computational Cost: This method inherits the high computational cost of TTRL while introducing additional computation. Are these additional overheads significant compared to the TTRL baseline?

---

> ### Author Response · Authors · 2025-11-26
> **Responses to Reviewer 7W3F (Part 1)**
>
> We thank the reviewer for the thoughtful and constructive feedback. We appreciate the time taken to review our work and the opportunity to clarify our contributions.
>
> **Q1. High Hyperparameter Sensitivity.**
>
> We thank the reviewer for this comment. We wish to clarify that RESTRAIN does not require extensive hyperparameter tuning. While our ablation study explores a wide parameter space for scientific completeness, the method performs robustly within a consistent training setting. The performance drops observed in the ablations reflect expected behavior when parameters are pushed to extremes, rather than inherent instability. Notably, we used a single set of hyperparameters across all benchmarks (Table 1), demonstrating that our approach generalizes effectively without the need for heavy, task-specific parameter search. We will clarify this in the revision to prevent any misunderstanding regarding the tuning overhead.
>
> **Table 1:** Hyperparameters used in our experiments.
>
> | **Hyperparameter** | **Symbol** | **Qwen3-4B-Base***(DAPO-14k)* | **OctoThinker-8B-Hybrid***(DAPO-14k)* | **Qwen3-4B-Base***(Synthetic)* |
> | :--- | :---: | :---: | :---: | :---: |
> | Mean of calculating pseudo-label weight | $\mu$ | 1.0 | 1.0 | 1.0 |
> | Bias of calculating pseudo-label weight | $\sigma$ | 0.5 | 0.5 | 0.5 |
> | Negative advantage offset | $\delta$ | 1.0 | 1.0 | 1.0 |
> | Majority size threshold | $\kappa$ | 3 | 3 | 3 |
> | Mean of calculating prompt weight | $\mu_{prompt}$ | 1.0 | 1.0 | 1.0 |
> | Bias of calculating prompt weight | $\sigma_{prompt}$ | 0.5 | 0.5 | 0.5 |
>
> **Q2. Limited Novelty**
>
> We thank the reviewer for this comment. We respectfully wish to clarify the nature of our contribution. Rather than an incremental adjustment to TTRL or a systems-level engineering effort, RESTRAIN introduces a fundamentally different paradigm for unsupervised reasoning. Specifically, it shifts the learning objective from a 'Winner-Takes-All' selection mechanism to a 'Distributional Self-Penalization' framework.
>
> This distinction is theoretical and algorithmic:
> 1. A Shift in Learning Paradigm:
> TTRL operates on the "Consistency Assumption" that the majority vote is the ground truth. It creates a binary learning signal that reinforces the majority, ignores the rest. RESTRAIN rejects this binary assumption. We introduce a probabilistic, distributional objective where the model learns from the entire spectrum of its outputs. By weighting prompts and pseudo-labels and applying negative advantages to low-confidence regions, we change the fundamental mechanism of how the model learns from unlabeled data.
> 2. Algorithmic Novelty:
> A systems-level engineering effort typically involves hyperparameter tuning, scaling infrastructure, or data filtering (like SRT’s easy prompt filtering). In contrast, RESTRAIN introduces a novel mathematical objective function, rather than relying on system engineering efforts, such as data filtering (SRT’s easy prompt filtering).
>     - Negative Rollout Penalization is to explicitly punish low-quality reasoning paths when consensus is weak, turning the absence of a clear signal into a learning opportunity.
>     - Prompt/pseudo-label Weighting is a derived confidence metric included directly in the loss computation to modulate gradient updates.
>     - These are algorithmic innovations that distinguish our work from engineering optimizations.
>
> 3. Solving Fundamental Failure:
> The difference in paradigm is evidenced by the training dynamics. TTRL suffers from model collapse because its "winner-takes-all" paradigm reinforces spurious errors. RESTRAIN’s "self-penalizing" paradigm inherently stabilizes training without external intervention.
>
> Additionally, we would like to note that RESTRAIN does not use curriculum learning. We do not add data in stages, change task order, or use progressive schedules, such as the "easy prompt" filtering seen in baselines like SRT.
>
> We will revise the manuscript to explicitly contrast these paradigms, ensuring the fundamental divergence from existing methods is clear.

---

> ### Author Response · Authors · 2025-11-26
> **Responses to Reviewer 7W3F (Part 2)**
>
> **Q3. Computational Cost**
>
> We thank the reviewer for raising the important question of computational cost. We are happy to clarify that RESTRAIN has nearly identical computational overhead to TTRL. Both methods share the same training hyperparameters—specifically rollout count, maximum sequence length, batch size, and number of epochs. Consequently, the most resource-intensive operations (LLM generation and policy forward/backward passes) are identical.
>
> RESTRAIN introduces only negligible overhead through lightweight operations on the generated rollouts: grouping unique answers, computing normalized pseudo-label weights, and applying consensus-gated offsets. These are simple operations that do not require additional model passes or parameters. Furthermore, the prompt-level weights are derived from a one-time offline computation, adding only a small constant setup cost rather than a recurring per-step burden.
>
> To validate this, we conducted a runtime analysis on the Qwen3-4B-Base model, which was trained on the DAPO-14k-MATH dataset (see Table 2). The results confirm that RESTRAIN maintains a training time per step comparable to both TTRL and standard gold-label training. While TTRL may exhibit shorter total training time due to early stopping caused by model collapse, the computational cost per step remains equivalent.
>
> We will revise the manuscript to include this discussion of computational cost in the appendix.
>
> **Table 2.** Computational Cost Analysis
>
> | Method | Training Time (Step 1) | Average Response Length |
> | :--- | :---: | :---: |
> | **Train with Gold Label** | 144s | 758.0 |
> | **TTRL** | 216s | 746.8 |
> | **RESTRAIN(Ours)** | 155s | 776.8 |

---

### Official Review · Reviewer_Fb6c · 2025-11-01

**Soundness:** 3
**Presentation:** 4
**Contribution:** 3
**Rating:** 6
**Confidence:** 4

**Summary:**

This paper proposes a new GRPO-based loss for self-training. Specifically, it does not rely solely on majority-voted answers but instead treats all answers as potentially correct, assigning different weights and penalizing low-consistency examples. Through extensive experiments on math and science datasets and two different models, the authors demonstrate that their self-training method outperforms previous approaches.

**Strengths:**

1. The proposed method is intuitive, well-motivated, and clearly described. It is simple yet effective.
2. The method achieves significant improvements over comparative self-training approaches across different models and two task domains.

**Weaknesses:**

1. The proposed loss and experiments are too closely tied to GRPO. It remains uncertain whether the method is compatible with other RL algorithms, such as PPO and PRIME, as compared by TTRL.
2. The tested datasets and models are relatively limited. Although the proposed method shows promising results in this paper, it is unclear whether it is biased toward inherently stronger models with certain specialized skills (both are base models), which may not generalize well in practice.

**Questions:**

1. How can TTRL be adapted to other RL framework and what are the results?
2. Could you provide results using Llama-3.1-8B-Instruct and Qwen2.5-Math-1.5B or 7B models?

---

> ### Author Response · Authors · 2025-11-26
> **Responses to Reviewer Fb6c**
>
> We are grateful for the reviewer’s positive assessment of our work and for the valuable suggestions to further improve the manuscript.
>
> **Q1. How can TTRL be adapted to other RL frameworks and what are the results?**
>
> We thank the reviewer for the question. Our approach is framework-agnostic: it only reshapes how per-trajectory losses are weighted and penalized, so it can sit on top of any policy-gradient framework. We used an action-resampling RL algorithm (GRPO) to keep the comparison controlled because all baselines we evaluate (including TTRL and SRT) are reported with the same family of algorithms; this isolates method-level effects from RL algorithm differences. Adapting to PPO is straightforward: replace the GRPO term with the standard PPO clipped surrogate and compute advantages with a value baseline.  We will add a brief appendix discussion that spells out the one-to-one substitution for PPO in our training loop.
>
> **Q2. The tested datasets and models are relatively limited.**
>
> We thank the reviewer for raising the important question regarding model generalization. To demonstrate that RESTRAIN is not biased toward specific base models, we highlight the diversity inherent in our original experimental design and provide new additional experiments on three different capacity models that confirm our method’s robustness across different model families and training stages.
> 1. Diversity in Existing Setup:
> Our original submission evaluates models on two models with two training datasets. We tested Qwen3-4B-Base (a standard base model) alongside OctoThinker Hybrid-8B-Base. It is crucial to note that OctoThinker is not a standard base model, but a specialized, highly optimized reasoning model derived from Llama-3.1-8B. This model underwent a specific "Stable-then-Decay" two-stage mid-training strategy designed to maximize reasoning capabilities. The fact that RESTRAIN significantly improves OctoThinker confirms our method is not limited to "weak" base models but can scale effectively on top of strong, specialized models.
> 2. New Experiments on different capacity Models and training sets:
> To additionally address the concern, we conducted experiments on three different capacity models: a base model: Qwen3-1.7B-Base, a math-specific model: Qwen2.5-math-7B, and an instruct model: Llama-3.1-8B-Instruct, cross two training datasets(DAPO-14k-math and NuminaMath-10k) in table 1. Consistent with our findings on base models, RESTRAIN outperforms the TTRL baseline under all settings. This confirms that our self-penalization mechanism is also effective for instruct models.
>
> By validating across Qwen (Base and math), OctoThinker (Specialized Mid-trained), and Llama (Instruct), we demonstrate that RESTRAIN generalizes across model families and training datasets. We will add these new results and the detailed characterization of the OctoThinker baseline to the revision to clarify the method's robustness.
>
> **Table 1** Additional results.
>
> | **Model** & Dataset | **Method** | **MATH500** | **AMC23** | **AIME24** | **AIME25** | **Olym.** | **Avg** |
> | :--- | :--- | :---: | :---: | :---: | :---: | :---: | :---: |
> | **Qwen2.5-Math-7B** | Base | 52.8 | 44.0 | 16.6 | 3.3 | 17.8 | 26.9 |
> | *(10k NuminaMath)* | Gold | 79.7 | 64.0 | 28.0 | 14.6 | 39.6 | 45.1 |
> | | TTRL | 77.9 | 61.5 | 18.0 | 8.0 | 37.2 | 40.5 |
> | | **RESTRAIN** | **79.3** | **62.5** | **26.7** | **13.9** | **40.7** | **44.6** |
> | | | | | | | | |
> | **Qwen3-1.7B-Base** | Base | 56.4 | 30.1 | 4.5 | 3.9 | 22.8 | 23.5 |
> | *(10k NuminaMath)* | Gold | 69.9 | 42.0 | 9.7 | 3.1 | 31.5 | 31.2 |
> | | TTRL | 63.2 | 36.2 | 5.4 | 3.5 | 26.0 | 26.8 |
> | | **RESTRAIN** | **66.9** | **41.0** | **8.3** | **5.2** | **29.6** | **30.2** |
> | | | | | | | | |
> | **Llama-3.1-8B-Instruct** | Base | 49.6 | 24.4 | 5.4 | 0.8 | 17.2 | 19.4 |
> | *(14k DAPO)* | Gold | 53.4 | 28.9 | 6.4 | 0.2 | 20.2 | 21.8 |
> | | TTRL | 47.5 | 25.0 | 6.1 | 0.8 | 17.3 | 19.3 |
> | | **RESTRAIN** | **51.2** | **26.4** | **7.1** | **1.1** | **17.3** | **20.6** |

---

### Meta-Review · Area_Chair_5dP6 · 2025-12-16

**Summary:**

All reviewer concerns have been responded to in the rebuttal. The authors have done a good job to address most of concerns (see my comment below), either via supplementary experiments or text explanations.

**Reviewer Concerns:**

Reviewer concerns AC thinks were addressed by the rebuttal:

Most of concerns have been addressed by the authors in the rebuttal. In particular,

1. The proposed loss and experiments are too closely tied to GRPO.

AC's comment: The authors claimed that the proposed method is framework-agnostic.

2. The tested datasets and models are relatively limited.

AC's comment: The supplementary experiments will alleviate reviewer's concern.

3. High hyperparameter sensitivity.

AC's comment: The authors replied that the performance drops observed in the ablations reflect expected behavior when parameters are pushed to extremes. The authors use constant hyper-parameter in the experiments. kappa = 3 is empirically stable.

Reviewer concerns AC believes are still outstanding:

1. Limited novelty.

AC's comment: Typically, it is hard to convince reviewers about the novelty of the paper in the rebuttal. AC is not 100% sure whether the response to the novelty concern can totally convince reviewers or not. AC would believe the rebuttal can mitigate the concern, but not fully.

**Reviewer Scores:**

In the first round of review, all reviewers vote for accept for this paper, and one reviewer champions the paper with a score of 8. Therefore, the paper is clearly above the accept bar of ICLR. The novelty concern worries the reviewers and AC. So AC recommends Accept (poster).

---

### Decision · Program_Chairs · 2026-01-26

Accept (Poster)